# Prediction of Cancer Proneness under Influence of X-rays with Four DNA Mutability and/or Three Cellular Proliferation Assays

**DOI:** 10.3390/cancers16183188

**Published:** 2024-09-18

**Authors:** Laura El Nachef, Larry Bodgi, Maxime Estavoyer, Simon Buré, Anne-Catherine Jallas, Adeline Granzotto, Juliette Restier-Verlet, Laurène Sonzogni, Joëlle Al-Choboq, Michel Bourguignon, Laurent Pujo-Menjouet, Nicolas Foray

**Affiliations:** 1INSERM U1296 Unit “Radiation: Defense, Health, Environment”, Centre Léon-Bérard, 69008 Lyon, France; laura.el-nachef@inserm.fr (L.E.N.); lb38@aub.edu.lb (L.B.); anne-catherinejallas@lyon.unicancer.fr (A.-C.J.); adeline.granzotto@inserm.fr (A.G.); juliette.restier--verlet@inserm.fr (J.R.-V.); laurene.sonzogni@inserm.fr (L.S.); joelle.al-choboq@inserm.fr (J.A.-C.); michel.bourguignon@inserm.fr (M.B.); 2Department of Radiation Oncology, American University of Beirut Medical Center, Beirut 1107-2020, Lebanon; 3Department of Anatomy, Cell Biology and Physiological Sciences, Faculty of Medicine, American University of Beirut, Beirut 1107-2020, Lebanon; 4Université Claude Bernard Lyon 1, CNRS, Ecole Centrale de Lyon, INSA Lyon, Université Jean Monnet, ICJ UMR5208, Inria, 69622 Villeurbanne, France; maxime.estavoyer@inria.fr (M.E.); simon.bure@insa-lyon.fr (S.B.); pujo@math.univ-lyon1.fr (L.P.-M.); 5Département de Biophysique et Médecine Nucléaire, Université Paris Saclay—Versailles St. Quentin-en-Yvelines, 78035 Versailles, France

**Keywords:** cancer proneness, hyper-recombination, cellular proliferation, G2/M arrest, excess of relative risk of cancer

## Abstract

**Simple Summary:**

By hypothesizing that molecular hyper-recombination and cellular proliferation are among the major features of multi-factorial cancer proneness, by using skin fibroblasts derived from eight major cancer syndromes, a significant correlation was found between the hyper-recombination rate quantified by plasmid assay, proliferation capacity assessed by flow cytometry, and excess of relative cancer risk (ERR). The product of the hyper-recombination rate and capacity of proliferation described a linear function of ERR.

**Abstract:**

**Context:** Although carcinogenesis is a multi-factorial process, the mutability and the capacity of cells to proliferate are among the major features of the cells that contribute together to the initiation and promotion steps of cancer formation. Particularly, mutability can be quantified by hyper-recombination rate assessed with specific plasmid assay, hypoxanthine-guanine phosphoribosyltransferase (HPRT) mutations frequency rate, or MRE11 nuclease activities. Cell proliferation can be assessed by flow cytometry by quantifying G2/M, G1 arrests, or global cellular evasion. Methods: All these assays were applied to skin untransformed fibroblasts derived from eight major cancer syndromes characterized by their excess of relative cancer risk (ERR). Results: Significant correlations with ERR were found between hyper-recombination assessed by the plasmid assay and G2/M arrest and described a third-degree polynomial ERR function and a sigmoidal ERR function, respectively. The product of the hyper-recombination rate and capacity of proliferation described a linear ERR function that permits one to better discriminate each cancer syndrome. Conclusions: Hyper-recombination and cell proliferation were found to obey differential equations that better highlight the intrinsic bases of cancer formation. Further investigations to verify their relevance for cancer proneness induced by exogenous agents are in progress.

## 1. Introduction

Cancer is one of the most common diseases worldwide. Each year, about ten million individuals are diagnosed with cancer. With about 2.7 million new cases per year, Europe accounts for a quarter of all cancer cases worldwide, while it represents less than 10% of the world population [1,2]. Since antiquity, cancer has been known to result from a multistep and complex biological process and the carcinogenic factors have been well summarized by Hanahan and Weinberg [3]. However, there is evidence that the relative contribution of each of these factors to the carcinogenesis process is unequal. For example, by documenting only the G2/M arrest in cytogenetics through the G2 radiosensitivity assay, the group of D. Scott has obtained a correlation with cancer proneness [4]. In fact, the precise mechanisms governing the formation and progression of cancer cells are still subjects of ongoing debate. From animal models, the pioneering work of Berenblum et al. [5,6] proposed a pivotal hypothesis: the *primum movens* of cancer would be likely to occur through two distinct steps—*initiation* occurring mostly at the molecular scale notably in response to oxidative stress and *promotion* occurring mostly at the cellular scale [3]. Subsequent research from Foulds has thereafter proposed a third step, *progression*, that describes the growth of the tumor in its microenvironment (i.e., events mostly occurring at the tissue scale) [3,7]. Chronologically, molecular investigations have paralleled the hypotheses of the initiation, promotion, and progression steps with the identification of some tumor suppressor genes, called *caretakers* which are responsible for genome surveillance, and *gatekeepers*, which are responsible for the control of the cell cycle checkpoints [3,8]. Hence, when caretakers are mutated, the initiation step results in the production and amplification of DNA mutation through a process called hyper-recombination [9]. When gatekeepers are mutated, the promotion step results in the stimulation of cellular proliferation, which amplifies also the number of mutations in daughter cells. More recently, some other oncogenes, called *landscapers,* may be responsible for tumor progression in its microenvironment [3,10,11,12]. However, it is noteworthy that such a summary may be not consensual and our recent epistemological review of the history of cancer models suggests that the definition of the terms “tumor suppressors”, “caretakers”, and “gatekeepers” may have changed through history [3,10,11,12].

From our recent review [3] and among a plethora of contributors, the mutability and the capacity of cells to proliferate are the two major features of cells that contribute together to the initiation and promotion of cancer formation. These features depend on numerous factors like the type of the tissue and the chronicity of the oxidative stress, whether endogenous or exogenous [11,12,13]. Furthermore, according to gene mutations, it is likely that both these features may contribute differently to cancer formation. Hence, the cancer risk/proneness may be quantified by developing molecular and cellular assays capable of independently assessing DNA mutability on the one hand and cell proliferation capacity on the other hand [11,12,13]. Do such specific assays exist?

Regarding DNA mutability, the major predictive assays are as follows:-Recombination plasmid assays that have contributed significantly to our understanding of DSB misrepair. In these assays, circular plasmids hold two different mutations of the same antibiotic resistance gene in two copies. The hyper-recombination process randomly occurs in all the plasmid DNA sequences, whether incubated in cell extracts (cell-free assays) or transfected in cells (cellular assays). Hyper-recombination leads to the production of several recombined plasmids including those holding a wild-type gene. For this subset of plasmids, the resistance to selection medium permits the survival of bacteria strains (cell-free assays) or cells (cellular assays) containing hyper-recombined plasmids. Such assays have permitted us to establish a correlation between the rate of hyper-recombination and cancer proneness [9,14,15,16,17]. It is noteworthy that such plasmids reflect a spontaneous hyper-recombination process. However, the use of extracts from cells exposed to a given stress (cell-free assays) or cells exposed to stress while seeded as monolayers (cellular assays) may reflect a stress-induced hyper-recombination.-The hypoxanthine-guanine phosphoribosyltransferase (HPRT) assay consists of assessing the DNA mutability of the *HPRT* gene after the exposure of cells to a given stress. HPRT is an enzyme involved in the purine salvage pathway, crucial for recycling hypoxanthine and guanine into nucleotides, which are essential for DNA and RNA synthesis. When HPRT is functional, it converts 6-thioguanine (6TG), a purine analog, into its toxic nucleotide forms (TGMP, TGDP, and TGTP). These nucleotides integrate DNA, causing breaks and cell death. By contrast, the cells that hold mutations in the *HPRT* sequence survive in the presence of the purine analog 6-thioguanine (6-TG), which permits the quantification of mutability. It is noteworthy that the HPRT assay was initially used to assess spontaneous DNA mutability. However, a given stress can be applied to the cells before proceeding to the selection medium step [18,19].-*Anti-MRE11* immunofluorescence assay that permits one to quantify hyper-recombination via the MRE11 nuclease activity. When associated with NBS1 and RAD50 proteins, MRE11 has a spontaneous nuclease activity. However, such activity can be inhibited by the phosphorylation of ATM that triggers the formation of nuclear MRE11 foci easily quantifiable by immunofluorescence. A quantitative correlation has been pointed out between the number of MRE11 foci reflecting nuclease activity and cancer proneness [16,20,21,22].-Many other assays can measure mutability specifically. This is notably the case of cytogenetics with chromosome aberrations as endpoints [23] or modified pulsed-field gel electrophoresis combined with Southern blotting [24]. However, such assays are too time-consuming to be applied to a large spectrum of cells.-With regard to the cell cycle control assays, three non-exclusive scenarios can occur: arrest in G1, G2/M, or even in S. Some specific techniques can be used like those assessing the G1/S transition (like BrdU/EdU incorporation assay), or those evaluating the G2/M arrest (like mitotic index assay). However, since it sorts cells as a function of DNA quantity, flow cytometry can reliably quantify all the cell subpopulations in each cell cycle phase [4,25,26,27].

By using a collection of fibroblast cell lines deriving from patients suffering from one of the eight major human genetic syndromes associated with cancer proneness (namely, ataxia telangiectasia (AT, *ATM^−^*^/*−*^ mutations), Nijmegen’s syndrome (NBS, *NBS1^−^*^/*−*^ mutations), neurofibromatosis type 1 (NF1, *NF1^+^*^/*−*^ mutations), Fanconi anemia (FANC, *FANC^+^*^/*−*^ mutations), Bloom syndrome (BLM, *BLM^+^*^/*−*^ mutations), retinoblastoma (RB, *Rb^+^*^/*−*^ mutations), and *BRCA1^+^*^/*−*^ and *BRCA2^+^*^/*−*^ mutations), we have assessed the mutability and control of cell cycle with the independent assays described above to investigate whether/how the resulting data can predict cancer risk quantified by the corresponding epidemiological excess of relative cancer risk (ERR) values taken from the literature. The aim of this study is, although aware that carcinogenesis is a multi-factorial process, to verify whether two major carcinogenesis factors, namely DNA mutability and cellular proliferation, are sufficiently predominant to reliably quantify cancer risk from skin fibroblasts expressed as ERR.

## 2. Materials and Methods

### 2.1. Cell Lines

In all the experiments performed in this study, the human untransformed cutaneous fibroblasts were cultured as monolayers in Gibco modified Eagle’s minimum medium (DMEM) (Thermo Fisher, Waltham, MA, USA), supplemented with 20% fetal calf serum, penicillin, and streptomycin. The cells were maintained in the plateau phase of growth (95–99% in G0/G1 at passages lower than 12) [28]. The genetic and cellular characteristics of the cutaneous fibroblasts used are detailed in Table 1. Two radioresistant cell lines (1BR3 and MRC5), originating from apparently healthy individuals, served as controls [28]. These cell lines were obtained from the European Collection of Authenticated Cell Cultures (ECACC, Health Security Agency, Porton Down, Salisbury, UK) under the references #90011801 and #05011802, respectively. Fibroblast cell lines from retinoblastoma (RB) (GM01142 and GM02718), Fanconi anemia A and C (GM00369 and GM16754, respectively), Bloom’s syndrome (BLM) (GM02520 and GM02048), Nijmegen Breakage syndrome (NBS) (GM07166), and ataxia telangiectasia (AT) (GM22690) were purchased from the Coriell Cell Repositories (Camden, NJ, USA). Lastly, fibroblast cell lines from neurofibromatosis type I (NF1) (Rackham 37), from *BRCA1^+^*^/*−*^ and *BRCA2^+^*^/*−*^ mutation familial syndromes (202CLB, 203CLB, and 201CLB, respectively), and AT (AT4BI) belong to the COPERNIC lab collection. The COPERNIC cell lines were obtained with all the sampling protocols approved by the national ethical committee. The resulting cells were declared to the Ministry of Research under the numbers DC2008-585, DC2011-1437, and DC2021-3957. The radiobiological data of the COPERNIC collection are protected under reference IDDN.FR.001.510017.000. D.P.2014.000.10300 [29,30]. The average doubling time of all the cell lines used in this study was 28 ± 4 h, except for the *ATM*-mutated cells that had a doubling time of 30 ± 1 h.

### 2.2. X-ray Irradiation

Irradiations were conducted using a 6 MeV photon medical irradiator (SL 15 Philips, Amsterdam, The Netherlands) at a dose rate of 6 Gy/min at the anti-cancer Centre Léon-Bérard (Lyon, France). The dosimetry features were certified by the Radiophysics Department of the Centre Léon-Bérard. A dose of 2 Gy was applied to mimic one session of standard radiotherapy [28].

### 2.3. Nuclear Extracts

The experimental protocols to obtain nuclear extracts were described elsewhere [28,31].

### 2.4. Hyper-Recombination Plasmid Assay

The construction of the pTPSN plasmid was detailed elsewhere [9]. Briefly, the pTPSN vector contains a *neo* gene used as a reporter for transfection selection and two mutant hygromycin resistance (*hyg*) genes in which a Hind III linker was introduced either at the unique PvuI site or at a SacII site (Figure 1). The percentage of *hyg^+^*/*neo^+^* cells was used as an endpoint to reflect intrachromosomal hyper-recombination. A total of 20 µg of circular pTPSN were transfected into five flasks containing 5 × 10^6^ cells each by using the calcium phosphate co-precipitation protocol followed with a glycerol shock [9,16]. The cells were thereafter incubated in a growth medium for 48 h. The selection of the transfected cells was performed by adding 0.5 mg/mL G418 geneticin (Sigma-Aldrich, St. Louis, MO, USA) directly to the medium. After about 20 days, the visible colonies were stained and counted. Because of the low transfection ratio generally observed with untransformed human fibroblasts ratio, the experiments were repeated to reach a total of 20 neoresistant colonies, at least (Figure 1) [16]. The rate of hyper-recombination, H, assessed with the pTPSN plasmid was calculated, as a percentage, as follows:(1)$$H(\%)=\frac{number\;of\;colonies\;Hyg+}{number\;of\;colonies\;Neo+}\times 100$$

### 2.5. Hypoxanthine-Phosphoribosyl Transferase (HPRT) Assay

After a 3-day amplification in flasks, the cells in the plateau phase of growth were counted using a hemocytometer (Kisker Biotech GmbH & Co, Steinfurt, Germany) and diluted to a predefined number of cells (500 or 1000 cells) in 10 cm Petri dishes with DMEM medium supplemented with penicillin, streptomycin, and 20% bovine serum (Figure 2). The cultures were maintained for 8 days. After this step, 2 μg/mL 6-thioguanine (6TG) (#A4882; Sigma-Aldrich) was added in one subset of dishes to determine the mutation frequency. Eight days later, the culture medium was removed. The cells were washed and stained with crystal violet solution (75% ethanol, 25% crystal violet, #HT90132, Sigma-Aldrich). After a last wash with water, the colonies were scored by eye (Figure 2). The frequency of *HPRT* mutations, F, was calculated, as a percentage, as follows [18]:(2)$$F\left(\%\right)=\frac{number\;of\;colonies\;after\;exposure\;to\;6TG}{number\;of\;colonies\;initially\;seeded\;without\;exposure\;to\;6TG}\times 100$$

### 2.6. Immunofluorescence Assay

The immunofluorescence protocol and nuclear foci scoring were performed as described elsewhere [28,31]. Briefly, the monoclonal anti-mouse anti-*MRE11* antibody (#56211) from QED Bioscience (San Diego, CA, USA) was used at a dilution of 1:100 at 37 °C for 1 h. Incubation with the anti-mouse fluorescein (FITC) secondary antibody was performed at a dilution of 1:100 at 37 °C for 20 min. The foci-scoring procedure was performed manually and has received certification under the CE mark and ISO-13485 quality management system norms [28,31]. Furthermore, the foci-scoring procedure was applied in the frame of the Soleau Envelop and patents (FR3017625 A1, FR3045071 A1, and EP3108252 A1). More than 50 nuclei were analyzed per experiment per post-irradiation time, with three independent replicates performed. The Gaussian distributions of the number of foci were controlled routinely for each condition. Inter-reader foci scoring revealed no significant difference, whether performed manually or by the computerized ImageJ v1.5 or Olympus foci-scoring software (v2.0) [28,31].

### 2.7. Nuclease Activity Assay

A total of 10 μg of the nuclear extract of each cell line mentioned in Table 1, along with the reaction buffer (Tris HCL pH 7.4 (#T5941; Sigma-Aldrich), MgCl_2_ (#203734; Sigma-Aldrich), Adenosine triphosphate (ATP) (#V703A, Promega, Charbonnières-les-bains, France), EDTA pH 8 (#E7889, Sigma-Aldrich), glycerol highly purity 99% (C3H8O3) (#EU3550, Euromedex, Souffelweyersheim, France) in distilled water), and plasmid pRR322 (#SD0041; ThermoFisher, Bourgoin-Jallieu, France), were incubated for 10 min at 37 °C (Figure 3). The mix was then re-incubated with 0.2% SDS (#71736; Sigma-Aldrich) and 5 mM EDTA (#E7889; Sigma-Aldrich) at pH 8 for 15 min at 37 °C. A 1% agarose gel (#15510027; Invitrogen, ThermoFisher Scientific, Waltham, MA, USA) was prepared with 1x TAE (#EU0202-A; Euromedex) and SYBR DNA gel stain (#593102; Invitrogen, ThermoFisher Scientific). The gel was visualized under UV using a ChemiDoc imaging system software version 2.4.0.03 (#733BR-4153, Bio-Rad Laboratories, Hercules, CA, USA), and the bands were analyzed and quantified using the Image Lab software version 6.1.0 build 7 (Bio-Rad Laboratories) (Figure 3).

### 2.8. Flow Cytometry—Cell Cycle Analysis

The cell cycle analysis assays involved staining DNA with a saturating amount of DNA-binding dye. The exact number of cells was fixed with a 70% ethanol solution, permeabilized, and stained with DAPI (#D21490; Thermofisher Scientific) that binds to the grooves of DNA in A-T-rich regions. The samples were acquired at a low flow rate with linear amplification. DAPI was excited in the UV range of light. It was analyzed by using the FACSDiva LSRFortessa, software version 9.0.19, Le Pont de Claix, France) to determine the cell cycle phases represented by the DNA histogram. The percentage of evasion E was calculated as follows:(3)$$E\left(\%\right)=\left(\%\;of\;cells\;in\;G2/M\;after\;irradiation-\%\;of\;unirradiated\;cells\;in\;G2/M\right)+\;\left(\%\;of\;cells\;in\;S\;after\;irradiation-\%\;of\;unirradiated\;cells\;in\;S\right)$$

Similarly, the percentage G of cells in G2/M was calculated as follows:(4)$$G\left(\%\right)=\left(\%\;of\;cells\;in\;G2/M\;after\;irradiation-\%\;of\;unirradiated\;cells\;in\;G2/M\right)$$

### 2.9. Statistical Analysis

Data were obtained from the indicated number of independent experiments and expressed as the mean ± standard error of the mean (SEM). Statistical analyses were performed using the PRISM software version 9.5.1 (GraphPad Software, San Diego, CA, USA) or Kaleidagraph version 4.5.4 (Synergy Software, Reading, PA, USA). The statistical inference of the formula parameters was made on Python v3.10.12 using the spicy package.

## 3. Results

### 3.1. Recombination Plasmid Assays

The recombination pTPSN plasmid assay with neomycin (reporter) and hygromycin (selector) resistance genes has been applied to the collection of cell lines representing the major cancer syndromes. By applying this assay to the radioresistant untransformed human fibroblasts, the hyper-recombination rate H was generally found to be lower than 10% [16]. Here, the average value obtained from the 1BR3 and MRC5 controls was 3.5 ± 2%, in good agreement with the literature [16] (Figure 4). The highest hyper-recombination rate was reached with the *ATM*-mutated fibroblasts with an average of 56 ± 8% from the AT4BI and GM022690 cells. When the H data were plotted against the corresponding excess of relative cancer risk (ERR), a specific shape of curve appeared combining a curvilinear up to ERR values close to 6–8 and an exponential curve for ERR values ranging from 8 to 10 (Figure 4A and Appendix A). A biological interpretation of this curve was proposed in the Section 4.

### 3.2. HPRT Assay

We applied the HPRT assay to the same collection of fibroblast cell lines. The HPRT frequency F of the untransformed human radioresistant fibroblast was found to be 10 ± 3%. The F values reached their maximum with the *NBS*-mutated fibroblasts (82 ± 10%) (no data with *ATM*-mutated cells were available). When the HPRT data were plotted against the corresponding excess of relative cancer risk (ERR), a curvilinear function appeared with a progressive increase in the *HPRT* mutation rate as a function of the ERR values (Figure 4B and Appendix A). Interestingly, when the hyper-recombination plasmid data were plotted against the corresponding *HPRT* mutation frequency data, a sigmoidal curve appeared with two thresholds, one reflecting the emergence of DNA mutations to reach the group of syndromes with ERR ranging from 2 to 8 and the other reflecting the cancer syndromes with ERR higher than 8 (Figure 4C). Again, a biological interpretation of this curve was proposed in the Section 4.

### 3.3. Nucleases Activity Assay

Hyper-recombination requires active nucleases to ensure the DNA strand exchange process. Hence, theoretically, high nuclease activity should reflect high rates of hyper-recombination. However, it is likely that the hyper-recombination process involves specific and not all DNA nucleases. Consequently, as a first step, we incubated a circular plasmid into nuclear extracts from the fibroblast cell lines of the collection described above to investigate the global nuclease activity. As mentioned in Materials and Methods, the residual plasmid was placed, after the step of incubation in nuclear extracts, into an electrophoresis gel and the smear reflecting the fraction of DNA released (FDR) was quantified in arbitrary units (Appendix A). As expected, the FDR derived from the radioresistant control cells was the lowest (21 ± 3%), suggesting a minimal but not nil DNA nuclease activity corresponding to a weak genomic instability. By contrast, the highest FDR values (61 ± 10%) deriving from the *ATM-*mutated cells corresponded to the highest DNA nuclease activities. By plotting the FDR against the ERR values, no significant correlation appeared between the two parameters, even if a general trend was observed: ERR roughly increased with FDR (Figure 5A and Appendix A).

### 3.4. MRE11 Foci Assay

As specified above, the nuclease activity assay assessed by gel was not specific to any DNA nuclease. MRE11 is an endonuclease whose role is crucial in the DNA repair and signaling pathway. Particularly, immunofluorescence assay permits one to quantify the nuclear foci formed by the MRE11 endonuclease when it is inactivated by its phosphorylation by ATM [21]. However, there is no correlation possible between spontaneous MRE11 foci and hyper-recombination, since the level of spontaneous MRE11 foci was too low for the cell lines tested [16]. Furthermore, a large subset of cancer syndromes tested here are associated with the absence of MRE11 foci or an unusual shape of MRE11 foci: this is the case of the BS, FANC, NBS, and ATM cells, in agreement with the literature [32] (Appendix A). The exposure to ionizing radiation permits one to establish the kinetics of radiation-induced MRE11 foci and to determine the post-irradiation time tested at which the MRE11 nuclease activity is maximal. As a first step, the maximal number of MRE11 (MRE11max) decreased while the ERR increased, suggesting that the MRE11 activity is inasmuch high as ERR is high (Figure 5B and Appendix A). The relationship between MRE11 and the hyper-recombination rate was found like the MRE11max vs. ERR while there were neither correlation between MRE11max and *HPRT* mutation frequency, nor between MRE11max and FDR, suggesting again that HPRT assay does not account for the hyper-recombination process only and that the nuclease assay does not reflect the MRE11 nuclease activity specifically (Figure 5C and Appendix A).

### 3.5. Cytometry Assay

While the previous assays investigated the mutability power of the fibroblasts tested, the cytometry assay permits one to assess the potential cellular proliferation and the results of the G1, S, and G2 checkpoint arrests in response to a given stress. We investigated by cytometry the cell cycle control after a reference dose of 2 Gy. The percentage of cellular evasion appeared as a curvilinear function for the ERR values lower than six followed by an exponential for the ERR values higher than six (Figure 6A). The percentage of G2 escape obeyed a sigmoidal function of ERR, suggesting that the cellular proliferation capacity may be a good sensor of ERR as well. Conversely, there was no significant correlation between the percentage of G1 escape and ERR (Figure 6C and Appendix A). Again, a biological interpretation of these curves was proposed in the Section 4.

### 3.6. Combination with Hyper-Recombination and Loss of Cell Cycle Control

Since cancer initiation and promotion were hypothesized to consist in the interplay between hyper-recombination and loss of cell cycle control, we investigated whether the product of the hyper-recombination rate H, assessed with the pTPSN plasmid assay, and the percentage of G2/M escape G, assessed by flow cytometry, can predict ERR better than each assay separately (Figure 7). The H × G product described a pseudolinear function of ERR with a very good correlation coefficient (r = 0.98). Again, a biological interpretation of this curve was proposed in the Section 4 (Figure 7).

## 4. Discussion

### 4.1. ERR as a Parameter Reflecting Cancer Proneness

What clinical endpoint can best reflect cancer risk/proneness? Epidemiology may provide the best compromise between statistical robustness and clinical description through the relative cancer risk (RR) or the excess of relative cancer risk (ERR) parameters. Indeed, RR and ERR are generally calculated for a given cancer syndrome from a large cohort. However, RR and ERR are also calculated for a given type of cancer. In this study, we have systematically reviewed from the literature data the ERR and RR values for each syndrome investigated, and the values chosen concern either the incidence of all the cancer types or the incidence of the most frequent cancer type. However, it must be stressed that all the syndromes investigated in this study are rare diseases. Their incidence ranges from 1/1000 (*BRCA1* mutations) to 1/100,000 (*ATM* mutations). Hence, while the risk of cancer associated with these syndromes is very high, their rarity may also make difficult a precise calculation of RR and ERR. For these reasons, we have integrated with our ERR data a systematic relative error range of 15% for the ERR values, which corresponds to the average relative error range observed in the literature ERR and RR data. Appendix A summarizes the references from which the ERR values were chosen. In this study, the following ERR values were taken as 0.4, 1, 3.2, 2.4, 1.4, 3.6, 7, and 9 for *BRCA1*, *BRCA2*, *Rb*, *BLM*, *FANC*, *NF1*, *NBS1,* and *ATM* mutations, [33,34,35,36,37,38,39,40,41] (Appendix A). Further investigations are obviously needed to better document these parameters.

### 4.2. One or Two Reliable Assay(s) to Predict Cancer Proneness?

The molecular process of carcinogenesis remains to be better understood. Particularly, the relative contribution to each carcinogenic factor, notably those cited in the hallmarks of cancer by Hanahan and Weinberg, should be documented [11,12,42]. Several different assays have been used to predict cancer proneness—while they have reflected directly or indirectly some major features of carcinogenesis, their specificity for one carcinogenesis step is still debated. Historically, the first examples of these assays were based on cytogenetics, chromosome aberrations, and G2/M radiosensitivity [32]. However, cytogenetic assays are too time-consuming, and it is practically difficult to apply them at a large scale. Since DNA mutations have been hypothesized to be the *primum* movens of the carcinogenesis process [3], the assays based on the DNA break misrepair may account for cancer proneness [43]. In this study, this is notably the case of the recombination pTPSN plasmid [9] and *HPRT* mutation frequency [18] assays that do not reflect the same features of hyper-recombination: these assays include notably a step of proliferation with the colony formation, which may complexify the prediction of the cancer risk. The absence of correlation between ERR and both MRE11 and global nuclease activity assays also demonstrated that the nuclease step alone is not predictive of cancer risk/proneness. Similarly, the percentages of cellular evasion, G2/M, or G1/S escape do not predict ERR equally, suggesting that one molecular assay may not be sufficient, alone, to predict cancer proneness reliably. Hence, that is why we also investigated the predictive power of the product of DNA mutation and cell cycle control endpoints, namely the H × G product. Indeed, the H × G product may reflect the DNA mutability associated specifically with a given cancer syndrome amplified by the capacity of cell proliferation also associated with the same syndrome. Hence, the H × G product may be considered as proportional to the amount of DNA mutation amplified in daughter cells. The use of H × G product may be inasmuch interesting as DNA mutability and the capacity of cell proliferation may contribute unequally to cancer proneness as evoked in the Section 1. However, further investigations and additional cell lines and/or cancer syndromes are needed to consolidate our findings.

### 4.3. Biological Interpretations of the Data Curves Linked to ERR

Since 2014, our research group has developed a mechanistic and biomathematical model of the individual response to ionizing radiation based on the radiation-induced (RI) nucleo-shuttling of the ATM protein, a major sensor of genotoxic stress (RIANS) [44]. Briefly, without irradiation, the ATM molecules formed homodimers. After any oxidative stress, ATM dimers monomerize and DNA double-strand breaks (DSBs) are induced. Cytoplasmic ATM monomers diffuse to the nucleus and phosphorylate the X variant of H2A histones at the site of the DSB by forming γH2AX foci visible by immunofluorescence. This recognition step is specific to DSB repaired by non-homologous end-joining (NHEJ), the most predominant DSB repair pathway in humans. Delayed RIANS is associated with RI toxicity (radiosensitivity), RI cancers (radiosusceptibility), or RI accelerated aging (radiodegeneration) [32,44]. The cause of delayed RIANS is generally the sequestration of ATM monomers in the cytoplasm by overexpressed ATM substrate proteins, called X-proteins, specific to each individual, tissue, or syndrome [32,44]. The radiobiological features of about twenty genetic syndromes have been already characterized in the frame of the RIANS model. Three groups of radiosensitivity syndromes have been defined as follows:-group I: radioresistance with a fast RIANS;-group II: radiosensitivity and cancer proneness or aging proneness with delayed RIANS and the existence of X-protein;-group III: hyper-radiosensitivity and very high risk of cancer or aging disease with no functional RIANS or gross DSB repair defect [44].

In addition to the groups of radiosensitivity described above, we have already proposed a complement of a RIANS model to explain carcinogenicity from a major principle, the duality of the DNA mutation formation (initiation step; involving the DSB repair proteins) and the loss of cell cycle checkpoint control (promotion step; involving cell cycle repair proteins like the CHK1 and CHK2 proteins involved in the G2 and G1 arrest, respectively) (Figure 8).

To avoid the confusion *with the groups of radiosensitivity, three categories of cancer syndromes were proposed [43]:*-category 1: Cancer syndromes caused by the mutations of the genes directly involved in DNA damage recognition or repair: the mutated gene products are also substrates of ATM and localize in the cytoplasm in which they form complexes with ATM monomers. While the mutated gene products lead to high mutability via their biological function in the nucleus, they delay the RIANS as cytoplasmic X-proteins and notably prevent the ATM-dependent phosphorylation of the CHK1 and CHK2 proteins: cellular proliferation is, therefore, facilitated. AT or NBS are representative examples of this category (Figure 8A).-category 2: Cancer syndromes caused by the mutations of the genes directly involved in the cell cycle checkpoint control: the mutated gene products are also substrates of ATM and localize in the cytoplasm in which they form complexes with ATM monomers. While the mutated gene products favor cellular proliferation via their biological function, they delay the RIANS as cytoplasmic X-proteins and notably prevent ATM-dependent DSB recognition, therefore favoring error-prone hyper-recombination: DNA mutability is exacerbated. RB and Li-Fraumeni (heterozygous *p53* mutations) are representative examples of this category (Figure 8B) [43,45].-category 3: Cancer syndromes caused by the mutations of genes whose role in DNA damage recognition and repair and in cell cycle checkpoint control remains unclear. The PROS syndromes are representative examples of this category [46] (Figure 8C).

By considering each category of cancer syndromes and each group of radiosensitivity syndromes, it appears clear that the interplay between the over-expressed X-proteins and the ATM monomers induced by radiation in cytoplasm and diffusing in the nucleus is crucial for the fate of the cells.

-In group I cells, the number of ATM monomers is very abundant and that of X-proteins is very low. Hence, in group I cells, the hyper-recombination process is possible but very limited and obeys H(ERR) = k_0_ × ERR: the product k_0_ × ERR can be considered as a basal spontaneous hyper-recombination whose intensity increases with genomic instability, here represented by ERR: the higher the ERR, the more intense the basal spontaneous hyper-recombination.-In group II cells, there is competition between the X-proteins and the produced ATM monomers. Up to ERR = ERR_max_/2, the ATM monomers limit the phenomenon of hyper-recombination (the slope of the H curve decreases, which produces a curvilinear shape) but such action becomes less and less possible since the number of X-proteins becomes higher while ERR increases(Figure 9). It produces syndromes associated with a moderate risk of cancer due to a limited hyper-recombination accompanied by a lack of cell cycle control (e.g., category 2 syndromes like *BRCA1* and *BRCA2* mutations, Figure 7). At ERR = ERR_max_/2, the influence of ATM monomers to prevent hyper-recombination is minimal: less ATM monomers diffuse in the nucleus, and therefore, more hyper-recombination occurs. When ERR > ERR_max_/2, the mutations of the X-proteins are more severe and may concern as well the activity of ATM kinase, which renders ATM monomers and X-proteins inefficient: H(ERR) tends to k_0_ × ERR again but from a level of basal spontaneous hyper-recombination which is very high already. This may correspond to a category 1 syndrome.-In group III, the number of ATM monomers is very low, or ATM monomers are completely inactive, and the number of X-proteins is also very low (homozygous mutations lead to a loss of function): there is no brake for the spontaneous recombination process and H obeys k_0_ × ERR (Figure 9). This is typically the case of category 1 syndromes.

The polynomial expression of H(ERR) deduced from data fits (Figure 4A) obeys, therefore, the following differential equations:(5)$$\frac{dH}{dERR}=k_{0}-k_{H}(ERR)$$
(6)$$k_{H}\left(ERR\right)=k_{1}\;ERR\;({ERR}_{max}-ERR)$$

*k_H_(ERR)* describes the “brake” of the spontaneous recombination caused by the ATM monomers diffusing in the nucleus.

The solution of Equation (5) is:(7)$$H\left(ERR\right)=\frac{k_{1}}{3}{ERR}^{3}-\frac{{k_{1}ERR}_{max}}{2}{ERR}^{2}+\frac{k_{1}{ERR}_{max}^{2}}{4}\;ERR$$
according to the data fit and with H(0) = 0 (8)
and ERR_max_ being the maximal value of ERR.

The numerical ERR_max_ value is deduced from the above equations:(9)$$k_{0}=\frac{k_{1}{ERR}_{max}^{2}}{4}$$

In this study, k_1_ = 0.9, ERR_max_ = 10.28, and k_0_ = 23.77.

The G2/M escape has been shown to be dependent on the unrepaired DNA damage [32]. Hence, the G2/M arrest is necessary a bound process, constrained by a maximal threshold number of unrepaired DSB that leads to an upper limit of the G2/M escape and a minimal threshold number of unrepaired DSB for which a significant G2/M escape is possible (Figure 10). Hence, the G2/M escape G is naturally described as a sigmoidal function, the solution of the following logistic differential equation, as follows:(10)$$\frac{dG}{dERR}=c_{0}G\left(1-\frac{G}{G_{max}}\right)$$
with its standard solution
(11)$$G\left(ERR\right)=\frac{G_{max}}{1+\left(\frac{G_{max}}{G_{min}}-1\right)e^{-c_{0}ERR}}$$

The positive constant $G_{max}$ is the maximum percentage of G2/M escape reached for a radiation reference dose of 2 Gy, and $G_{min}$ is the percentage of G2/M escape with ERR=0. Numerically, in this study, we determined the value of the parameters $c_{0}=0.5473$, $G_{max}=10.98$, and $G_{min}=0.8655$.

From Equations (8) and (9) described above, the product (H × G) can be defined as follows:(12)$$\left(H\times G\right)\left(ERR\right)={k_{1}G}_{max}\frac{\frac{{ERR}^{3}}{3}-\frac{{ERR}_{max}{ERR}^{2}}{2}+\frac{{ERR}_{max}^{2}\;ERR}{4}\;}{1+\left(\frac{G_{max}}{G_{min}}-1\right)e^{-c_{0}ERR}}$$

One of the advantages of the H × G product is that it permits one to discriminate all the syndromes on a linear scale by taking into account hyper-recombination and G2/M arrest independently. Further investigations are needed to better document and justify such an approach.

Lastly, we examined the differences between radiosensitivity and cancer-proneness with such novel endpoints. The H × G product was plotted against the corresponding surviving fraction at 2 (SF2) that reflects radiosensitivity. A sigmoid function did appear, suggesting that the higher the radiosensitivity the larger the H × G product (Figure 11). Hence, by acting independently, hyper-recombination and the lack of cell cycle control converge together to an amplification of errors in cells that leads the process up to the frontier between promotion (cellular scale?) and progression (tissue scale). This is the limit of the validity of our model. A similar approach should be applied to investigate the prediction of the formation of cancer induced by a given carcinogenic factor.

## 5. Conclusions

The mutability related to hyper-recombination and the capacity of cells to proliferate are two major features of cells that may contribute together to the initiation and the promotion step of cancer formation. Here, we hypothesized that these last two features may be sufficiently important to predict spontaneous cancer risk (quantified by ERR) from the major cancer syndromes. A strong correlation was found between each of these features and ERR described by a 3rd-degree polynomial function and a sigmoidal function, respectively. The product of the hyper-recombination rate and capacity of proliferation described a linear function of ERR and may discriminate better the different cancer syndromes. Obviously, this study is limited by the number of cell lines investigated, the number of syndromes tested, the fact that we used only one type of cell (fibroblasts), and the fact that other physical and chemical agents may induce carcinogenesis. Further investigations are, therefore, required with new cell lines, genetic syndromes associated with cancer, new tissues, and new carcinogenic agents to consolidate our approach that already suggests that, although a multi-factorial process, cancer proneness can be reliably predicted by the quantification of hyper-recombination and the capacity of cells to proliferate.

## Figures and Tables

**Figure 1 cancers-16-03188-f001:**
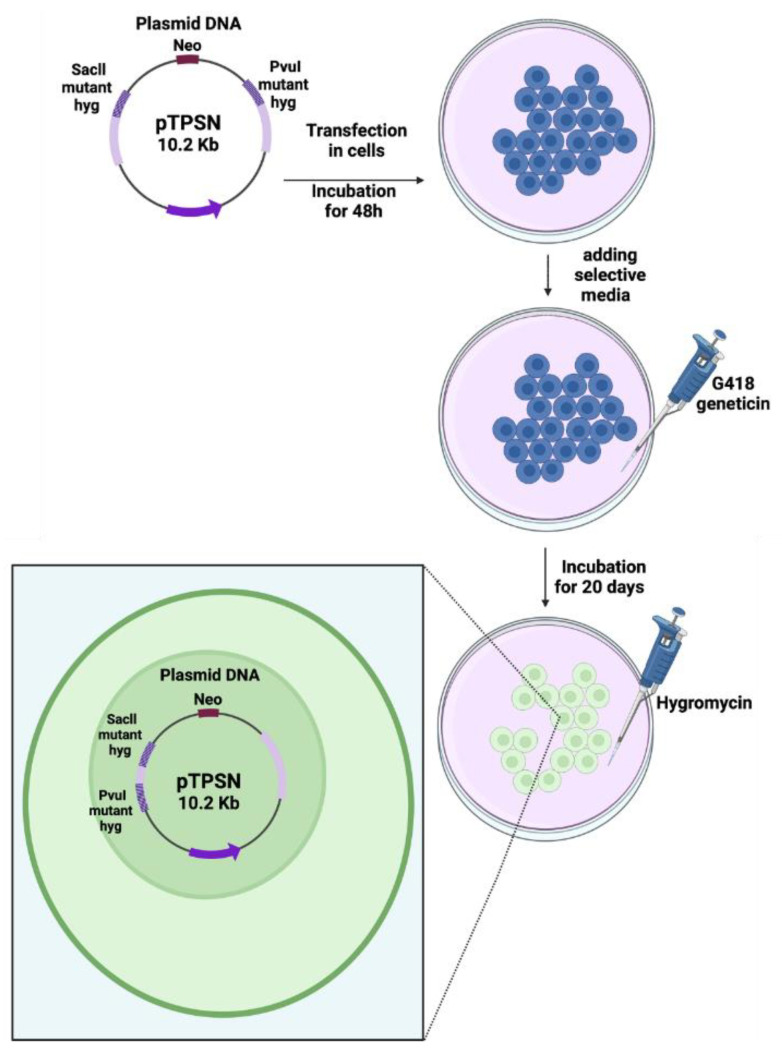
Schematic illustration of the recombination plasmid assay. The pTPSN plasmid was transfected in cells with the calcium phosphate method. After a 3-day culture, geneticin (neomycin) was added to select the cells containing the plasmid. After a 20-day culture, hygromycin was added to select the colonies of cells containing the hyper-recombined plasmid holding a wild-type copy of the hygromycin resistance gene [16].

**Figure 2 cancers-16-03188-f002:**
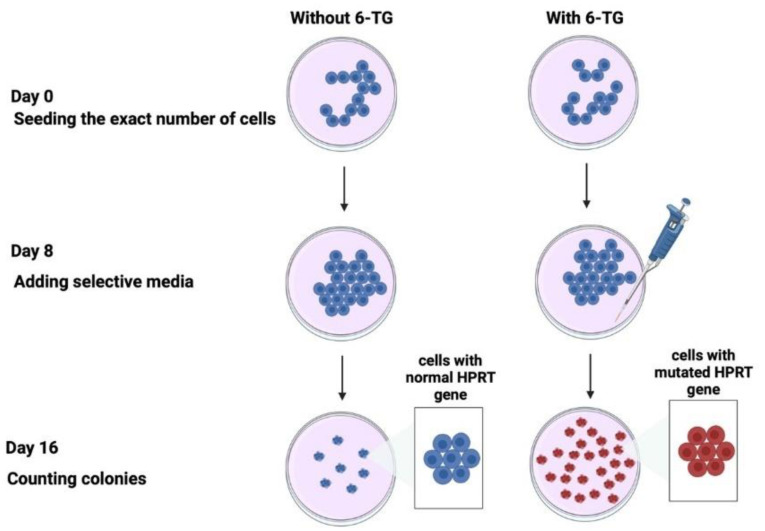
Schematic illustration of the HPRT assay. Two parallel subsets of cells were seeded with or without 6-TG. After 8 days of culture, the selective media was added or not to the cells. The cells with wild-type (non-mutated) *HPRT* genes do not survive in the culture; only the cells with non-functional (mutated) HPRT genes survive in the culture [18].

**Figure 3 cancers-16-03188-f003:**
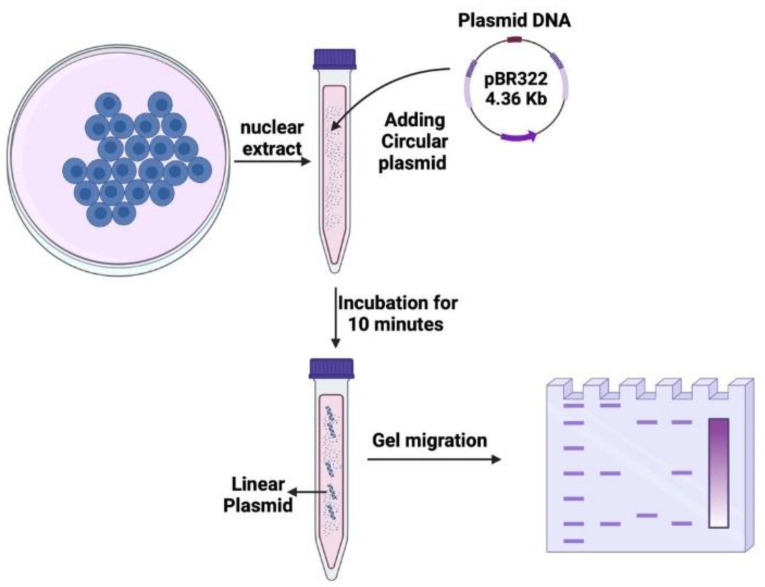
Schematic illustration of the nuclease activity assay. Circular plasmid DNA was incubated with nuclear proteins extracted from cells for 20 min. Following the incubation, the DNA samples were subjected to electrophoresis on agarose gel. The resulting gel displayed different bands and smear patterns, indicating varying degrees of nuclease activity.

**Figure 4 cancers-16-03188-f004:**
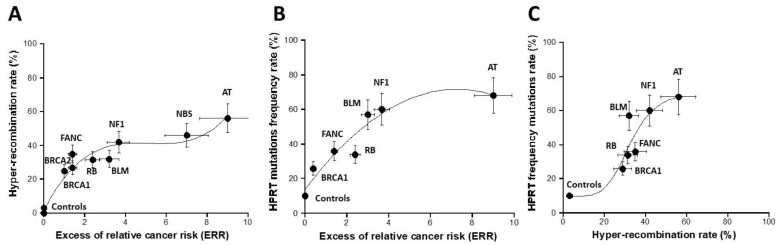
Assessment of hyper-recombination. The rate of hyper-recombination assessed with the pTPSN plasmid (**A**) and HRPT (**B**) assays plotted as a function of the corresponding ERR for each syndrome investigated. Each plot represents the mean ± standard error of the mean (SEM) of at least three independent replicates. Panel C represents the crossed correlation between the data shown in both A and B panels. The data were fitted to (**A**): a 3rd-degree polynomial function (y = 1.65 + 23.58x − 4.63x^2^ + 0.3x^3^; r = 0.979); (**B**) a 2nd-degree polynomial function (y = 13.8 + 15.95x − 1.1x^2^; r = 0.955); (**C**) a sigmoidal function. In all these experiments, each syndrome is represented by one representative fibroblast cell line: AT: AT4BI; NBS: GM07166; NF1: Rackham 37; BLM: GM02520; FANC: GM00369; RB: GM02718; BRCA1: 202CLB; BRCA2: 201CLB; controls: 1BR3 and MRC5. See the numerical values in Appendix A.

**Figure 5 cancers-16-03188-f005:**
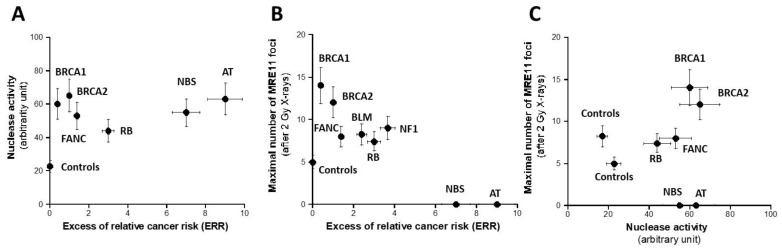
Assessment of nuclease activity. The rate of nuclease activity assessed by gel electrophoresis (**A**) and the maximal number of MRE11 foci assessed after 2 Gy X-rays (**B**) plotted as a function of the corresponding ERR for each syndrome investigated. Each plot represents the mean ± standard error of the mean (SEM) of at least three independent replicates. Panel (**C**) represents the crossed correlation between the data shown in both A and B panels. In all these experiments, each syndrome is represented by one representative fibroblast cell line: AT: AT4BI; NBS: GM07166; NF1: Rackham 37; BLM: GM02520; FANC: GM00369; RB: GM02718; BRCA1: 202CLB; BRCA2: 201CLB; controls: 1BR3 and MRC5. See the numerical values in Appendix A.

**Figure 6 cancers-16-03188-f006:**
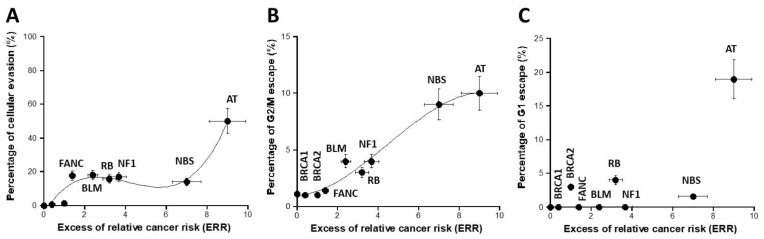
Assessment of cell cycle escape. The percentage of cellular evasion (**A**), G2/M escape (**B**), and G1 escape (**C**) were plotted as a function of the corresponding ERR for each syndrome investigated. Each plot represents the mean ± standard error of the mean (SEM) of at least three independent replicates. The data were fitted to (**A**): a 3rd-degree polynomial function (y = −4.2 + 19.88x − 5.38x^2^ + 0.43x^3^; r = 0.966); (**B**) a sigmoidal or a 3rd-degree polynomial function (y = 10.98/(1 + 11.69 × exp(−0.54 × x)); r = 0.986). In all these experiments, each syndrome is represented by one representative fibroblast cell line: AT: AT4BI; NBS: GM07166; NF1: Rackham 37; BLM: GM02520; FANC: GM00369; RB: GM02718; BRCA1: 202CLB; BRCA2: 201CLB; controls: 1BR3 and MRC5. See the numerical values in Appendix A.

**Figure 7 cancers-16-03188-f007:**
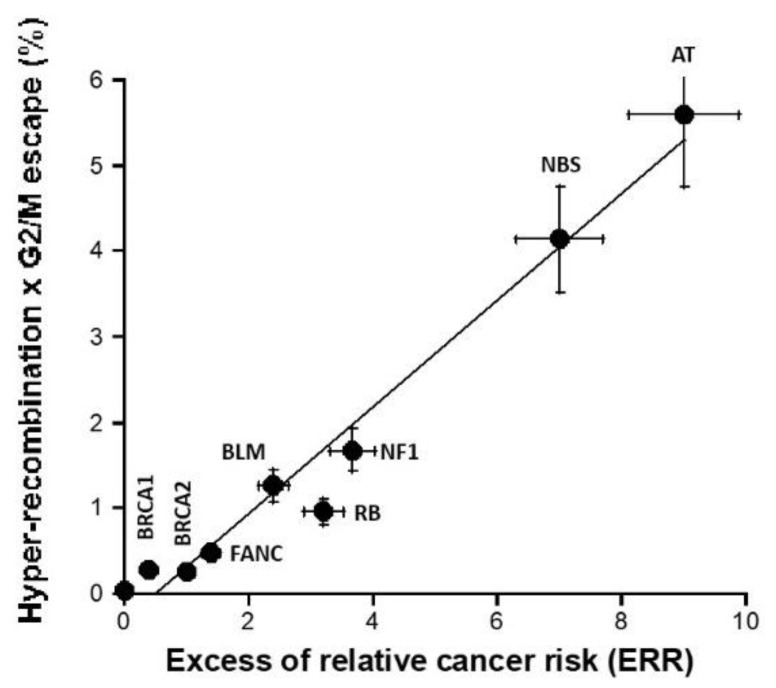
H × G product vs. ERR. The product (expressed in percentage) of the hyper-recombination rate H (assessed with the pTPSN plasmid assay and whose data are shown in Figure 4A) and the percentage of G2/M escape G (assessed by cytometry and whose data are shown in Figure 6B) was plotted as a function of the corresponding ERR for each syndrome investigated. Each plot represents the mean ± standard error of the mean (SEM) of at least three independent replicates. The data were fitted to a linear function (y= −0.3 + 0.623x; r = 0.983). In all these experiments, each syndrome is represented by one representative fibroblast cell line: AT: AT4BI; NBS: GM07166; NF1: Rackham 37; BLM: GM02520; FANC: GM00369; RB: GM02718; BRCA1: 202CLB; BRCA2: 201CLB; controls: 1BR3 and MRC5. See the numerical values in Appendix A.

**Figure 8 cancers-16-03188-f008:**
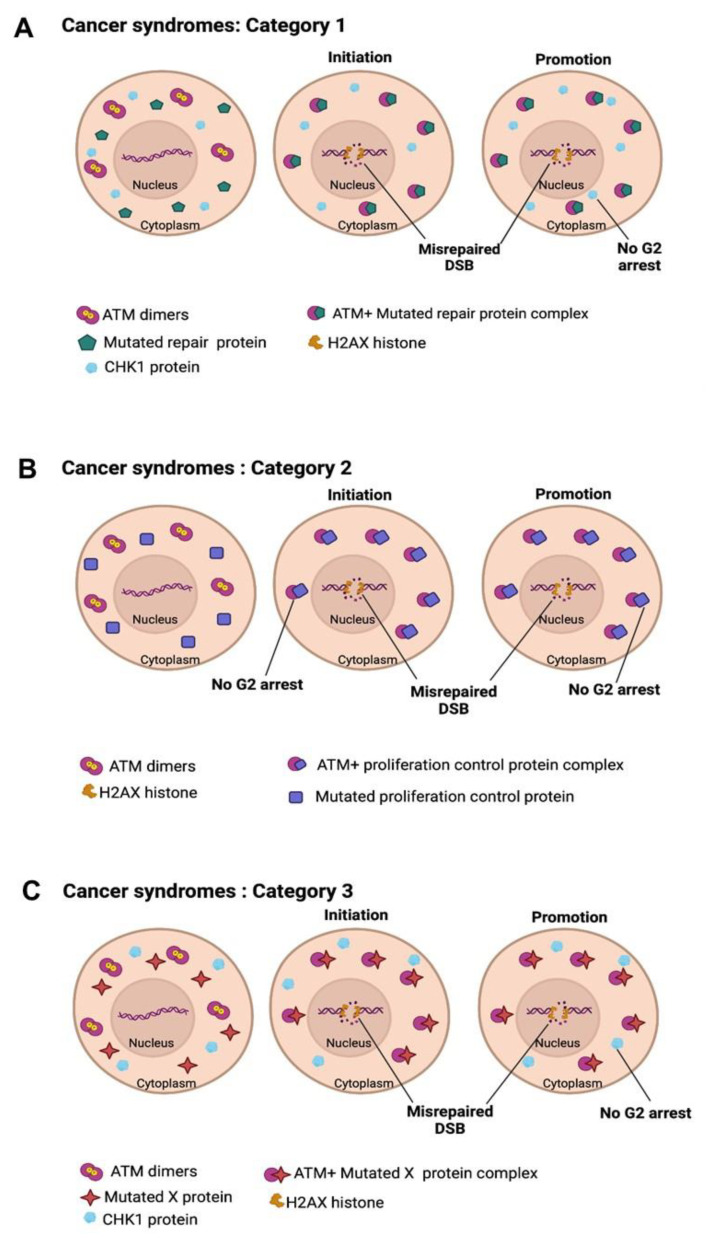
Schematic illustration of the carcinogenesis model explained by the RIANS model. (**A**) In category 1 cells, DNA repair proteins are mutated, and mutations are spontaneously generated. ATM monomers are sequestered by the cytoplasmic forms of the mutated proteins by forming ATM-X complexes. This sequestration prevents the phosphorylation of CHK1 and CHK2 proteins by ATM, which impairs the cell cycle checkpoints in G2/M and G0/G1, respectively. (**B**) In category 2 cells, cell cycle checkpoint control proteins are mutated. ATM monomers are sequestered by the cytoplasmic forms of the mutated proteins by forming ATM-X complexes. Such delay in RIANS promotes hyper-recombination cell proliferation and causes misrepaired DSB, leading to hyper-recombination in cells. (**C**) In category 3 cells, the mutated proteins are apparently not involved in DNA repair nor in cell cycle control. However, the sequestration of ATM by the cytoplasmic forms of X-proteins is sufficient to promote both hyper-recombination and cell proliferation.

**Figure 9 cancers-16-03188-f009:**
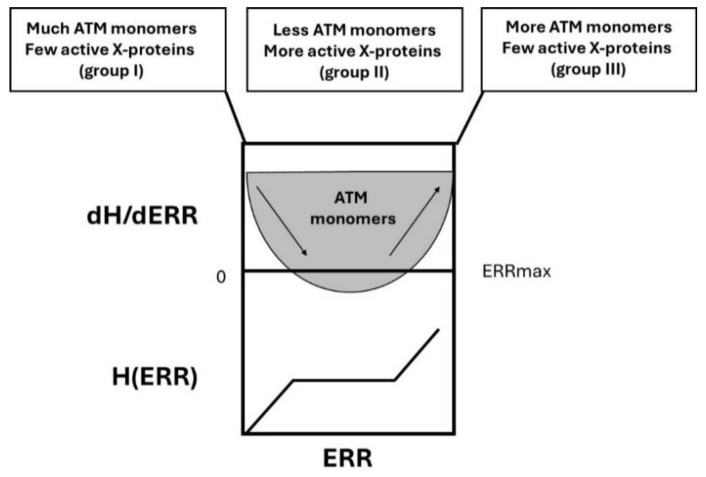
Schematic illustration of the H(ERR) factor and its derivative. In group I cells, with a low risk of cancer, the ERR value is close to 1, and there are few X-proteins and much more ATM monomers: hyper-recombination is less probable. In group II cells, with a high risk of cancer, the ERR value ranges from 2 to 6, and there are abundant X-proteins and much less ATM monomers that diffuse in the nucleus: hyper-recombination is highly probable. In group III cells, with a very high risk of cancer, the ERR value is larger than 6, and there are few X-proteins due to gene mutations but also few active ATM monomers despite their number: hyper-recombination is, therefore, very highly probable.

**Figure 10 cancers-16-03188-f010:**
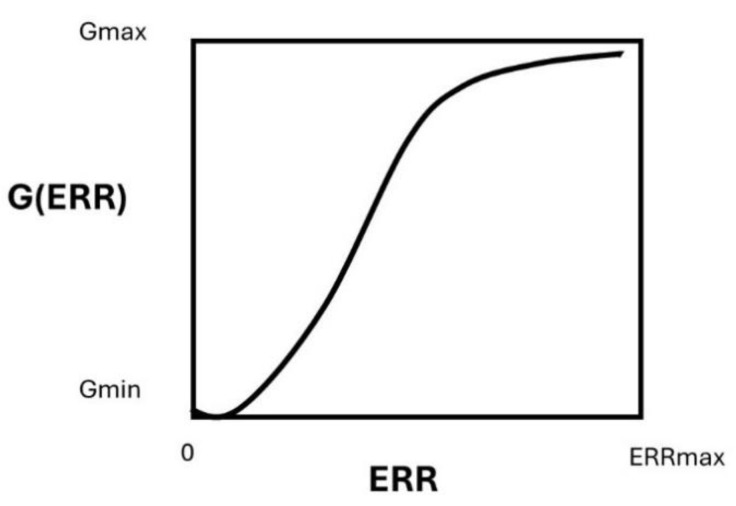
Schematic illustration of the G(ERR) factor.

**Figure 11 cancers-16-03188-f011:**
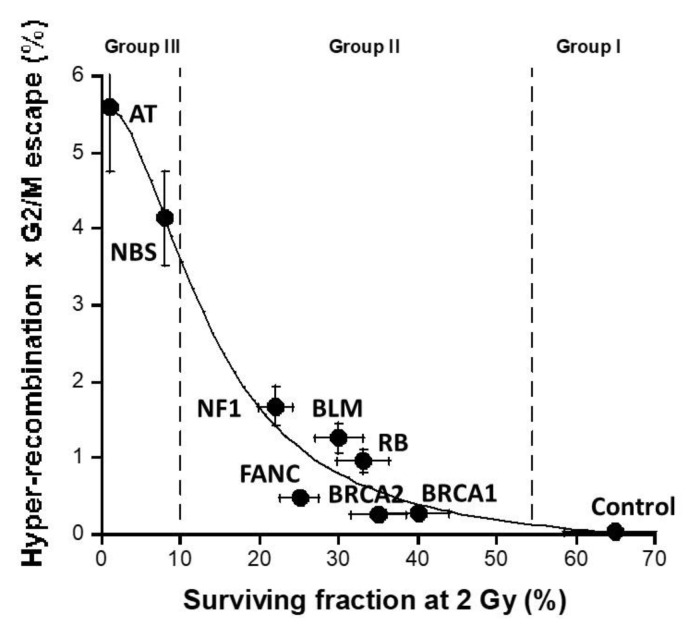
Relationship between the H × G product and the cellular radiosensitivity: the values of the H × G product shown in Figure 9 were plotted against the SF2 values for each syndrome [43]. The data were fitted by a sigmoidal function (r = 0.98).

**Table 1 cancers-16-03188-t001:** Major genetic and clinical features of the untransformed and transformed cell lines used in this study.

Cell Lines	Origin	Syndromes	Genetic and Clinical Features
1BR3	ECACC	-	Apparently healthy
MRC5	ECACC	-	Apparently healthy
AT4BI	COPERNIC	Ataxia telangiectasia	*ATM^−^*^/*−*^ mutation
GM22690	CORIELL	Ataxia telangiectasia	*ATM^−^*^/*−*^ mutation
201 CLB	COPERNIC	BRCA2	*BRCA2^+^*^/*−*^ mutation
202 CLB	COPERNIC	BRCA1	*BRCA1^+^*^/*−*^ mutation
203 CLB	COPERNIC	BRCA1	*BRCA1^+^*^/*−*^ mutation
GM01142	CORIELL	Retinoblastoma	*Rb^−^*^/*−*^ mutation
GM02718	CORIELL	Retinoblastoma	*Rb^−^*^/*−*^ mutation
Rackham 37	COPERNIC	Neurofibromatosis type 1	*NF1^+^*^/*−*^ mutation
GM00369	CORIELL	Fanconi anemia A	*FANCA^+^*^/*−*^ mutation
GM16754	CORIELL	Fanconi anemia C	*FANCC^+^*^/*−*^ mutation
GM02520	CORIELL	Bloom syndrome	*BLM*/*RECQL3^−^*^/*−*^ mutation
GM02548	CORIELL	Bloom syndrome	*BLM*/*RECQL3^−^*^/*−*^ mutation
GM07166	CORIELL	Nijmegen breakage syndrome	*NBS1^−^*^/*−*^ mutation

## Data Availability

All the data can be provided upon reasonable request.

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
