# Peer review of "Prediction of Cancer Proneness under Influence of X-rays with Four DNA Mutability and/or Three Cellular Proliferation Assays"

_cancers, 2024, doi:10.3390/cancers16183188_

Round 1

Reviewer 1 Report

Comments and Suggestions for Authors

1) The current abstract is a general fact. They should revise it by adding numerical results. At least half of the abstract should contain the authors' findings.

2)      The aims of the work should be added to the end of the introduction.

3)       Graphical abstract is required at the end of the introduction.  (In the graphical abstract, the methods and results should be represented as graphics)

4)      A lot of the cancers have an epigenetic reason. E.g. DNA methylation. These kinds of aberrations cannot be predicted from DNA mutagenesis assays. How do the authors justify this?

5)      The purpose of pTPSN plasmid assay in this work should be described in Methods or results.

6)      How the authors calculate Excess of relative cancer risk (ERR) (fig 4 x-axes)

7)      Mutagenesis may increase ROS levels. Why the authors did not measure it?

8)      Although the discussion is pretty long, but it is a restatement of the results. Their results have not been compared with previous data/results in discussion.

9)      The authors mentioned flow cytometry data for cell cycle control. However, FACS data is not available in the results parts. 

Author Response

Reviewer 1 :

We thank the reviewer 1 for his/her comments

  1. The current abstract is a general fact. They should revise it by adding numerical results. At least half of the abstract should contain the authors' findings.

OK. In the first version of the abstract, half of the text was the description of our findings. However, we are aware that our results are not necessarily numerical since we investigated the existence of correlations, and we described the form of the mathematical functions obtained. See modified text

  1. The aims of the work should be added to the end of the introduction.

Ok, see modified text at the end of Introduction

  1. Graphical abstract is required at the end of the introduction.  (In the graphical abstract, the methods and results should be represented as graphics)

Ok, see graphical abstract

  1. A lot of the cancers have an epigenetic reason. E.g. DNA methylation. These kinds of aberrations cannot be predicted from DNA mutagenesis assays. How do the authors justify this?

OK We agree that DNA methylation is involved in cancer but literature and our recent review remain unclear on two major points: 1) is there any a causal and quantitative relationship between cancer and DNA methylation? To this question, the reply is unclear since it is unclear whether the most reliable endpoints is hypomethylation, hypermethylation of one gene or all the genome. 2) it is unclear whether DNA methylation impairments are not also associated with DNA mutations of genes involved in genome integrity and cellular proliferation control. By contrast, aware that cancer is a multi-factorial process, our data showed that there is a correlation with our markers (hyper-recombination and proliferation) and cancer proneness independently of DNA methylation. See modified text in abstract and Discussion.

  1. The purpose of pTPSN plasmid assay in this work should be described in Methods or results

OK in fact the 3rd reviewer asks some details of the major principles of these assays in Introduction for broader audiences See modified text in Introduction

      6.How the authors calculate Excess of relative cancer risk (ERR) (fig 4 x-axes)

OK We have explained how we have chosen the ERR from literature in Disucssion. In fact we transposed a new paragraph in Materials and methods. See modified text

  1. Mutagenesis may increase ROS levels. Why did the authors not measure it?

OK. Mutagenesis may increase ROS levels and reciprocally. However, some ROS may not be the direct cause of cancer : we preferred therefore to apply functional assays approach than simple biochemical approach : in other terms, our assays will be more informative and more specific than a simple measurement of ROS level.

  1. The authors mentioned flow cytometry data for cell cycle control. However, FACS data is not available in the results parts. 

 OK. The cytometry data are present as % Escape, %G1 and %G2 arrest. To reach the requirement of one reviewer, we added a table in supplementary data with all the numerical data. See table S1 that summarized all the numerical values.

Reviewer 2 Report

Comments and Suggestions for Authors

The study “Prediction of cancer proneness with DNA mutagenicity and/or cellular proliferation assays” is devoted to elaboration of a simple approach for determination of individual cancer proneness. Authors used fibroblast primary cultures of patients with inheritable predisposition to cancer analyzing the levels of (1) recombination, (2) gene mutations in the hypoxanthine phosphoribosyl transferase (HPRT) gene and (3) activity of cell proliferation under influence of X-ray radiation.

Major comments

1.     The fact that only X-ray radiation is used in the study as carcinogenic factor definitely limits its versatility, that is, its application to many other carcinogenic factors, both domestic and professional. 30% of cancer is associated with exposure to chemical carcinogens and the authors did not analyze chemical carcinogens at all. Thus, the application of this approach is limited to the assessment of sensitivity to radiation carcinogenesis.

2.     The Cancer Genome Atlas program showed huge number of genes involved, regulating not only recombination and cell proliferation, that in concordance with the publications of Professors Hanahan and Weinberg perfectly describing hallmarks of cancer. Thus, the information, which could be get using proposed approach is rather limited, also some correlation between cancer proneness and the intensity of DNA repair via homologous recombination and activity of cell proliferation may be observed.

3.     The data revealing the association between cancer proneness and the intensity of DNA repair via homologous recombination and activity of cell proliferation should be presented more precisely. In the Result section authors demonstrate only some samples of the analyzed curves without detailed description how they were processed and integral data obtained. It is necessary to add tables including all the primary cell cultures analyzed, which could demonstrate final results of every analysis performed, so that to make possible to see the associations.

4.     There is one more limit of the approach proposed in the study: it is performed using the cell of only one histogenesis type and radiation caused mainly blood malignancies, thus, in the analysis of cancer proneness, it is necessary to use cells of different histogenesis.

5.     There are many terminological and conceptual inaccuracies in the text of the manuscript, which are described in minor comments.

Minor comments:

1.     Mutagenicity is a property of genotoxic agents/factors to induce DNA alterations, thus, it should not be used as a property of DNA. In the title and in the text “DNA mutagenicity” should be corrected to “DNA mutability”.

2.     The phrase “Cancer proneness is associated with molecular hyper-recombination and cellular proliferation” should be corrected as Cancer proneness is associated with many factors, the line of which is very wide, and here it sounds as hyper-recombination and cellular proliferation are the only properties determining cancer proneness. In the context of the proposed methodology the idea of the phrase should be alike this one: “Activities of recombination and cellular proliferation are associated with radiation induced cancer proneness”.

3.     The phrase “etiology of cancer remains poorly understood” is not correct, nowadays carcinogenic factors are well known and key characteristics of carcinogens are well described in the documents of the International Agency for Research on Cancer, Lyon, WHO. This agency elaborated perfect classification of carcinogenic factor of different nature (chemical, biological agents and physical factors). Modern concept of carcinogenesis is well-presented in the whole number of studies, showing simultaneous influence of multiple genotoxic and non-genotoxic promoting factors both on tumor appearance and tumor progression. Hallmarks of cancer, presenting the main peculiar characteristic of cancer cells, are also well-presented in the works of Prof. Weinberg and Prof. Hanahan.

4.     The phrase “Molecular investigations have paralleled the hypotheses of the initiation, promotion and progression steps with the identification of some tumor suppressor genes, called caretakers when responsible for genome surveillance, and gatekeepers, when responsible for control of the cell cycle checkpoints” requires serious revision. Initiation, promotion and progression are the steps of carcinogenesis, which are well documented, it is not the hypothesis, concerning the genes, involved in carcinogenesis. Exposure of cells to genotoxic compounds, leading to mutation appearance in many genes involved in carcinogenesis, corresponds to cancer initiation, and regulation of these gene activity by non-genotoxic carcinogens, corresponds to the process of promotion. As concerns the genes involved into carcinogenesis, they may be separated into different groups in dependence on their function. Thus, in parallel may be considered classifications of the genes involved into carcinogenesis to (1) oncogenes and genes of tumor suppressors or (2) caretakers and gatekeepers. Although, nowadays, the publications of Hanahan and Weinberg determined more popular classification of the genes, involved into carcinogenesis, dependently of their involvement in the different processes corresponding to Hallmarks of cancer. Cancer progression occurs as the result of the influence of both genotoxic and non-genotoxic agents/factors. A very interesting results was obtained by Atlas of cancer genome program showing that only a small number of genes are not involved in carcinogenesis.

5.     Proneness to cancer depends on the enzyme system of carcinogen activation and detoxification and all the enzymes of DNA repair, antitumor immunity, epigenetic regulation and other types of DNA transcription activation. Complex composition of polymorphic variants of all these gene, which function under influence of different environmental factors, determine the individual risk of cancer (or cancer proneness). Is it reasonable to determine just two factors influenced the process, even if these factors are important? At least, t is necessary to present the goal of the study focusing on carcinogenesis induced by radiation. It is also very important, in particular, for people working with professional radiation exposure and for patients treated with radiotherapy.

6.     The phrase “With regard to cell cycle control assays, three major categories exist: those assessing the G1/S transition (like BrdU/EdU incorporation assay), those evaluating the G2/M arrest (like mitotic index assay), and finally flow cytometry that encompasses all the stages of cell cycle” requires more precise explanation.

7.     The authors pointed out that “More than 50 nuclei were analyzed per experiment per post-irradiation time, with three independent replicates performed”. It should be pointed out how it corresponds to OECD guidelines.

8.     The data obtained should be presented in more details to see what syndrome causes what change of the property, please, provide the corresponding information in details in tables in the main text.

Comments on the Quality of English Language

Use of English terminology should be improved.

Author Response

Reviewer 2 :

We thank the reviewer 2 for his/her comments

The study “Prediction of cancer proneness with DNA mutagenicity and/or cellular proliferation assays” is devoted to elaboration of a simple approach for determination of individual cancer proneness. Authors used fibroblast primary cultures of patients with inheritable predisposition to cancer analyzing the levels of (1) recombination, (2) gene mutations in the hypoxanthine phosphoribosyl transferase (HPRT) gene and (3) activity of cell proliferation under influence of X-ray radiation.

  • Major comments :

  1. The fact that only X-ray radiation is used in the study as carcinogenic factor definitely limits its versatility, that is, its application to many other carcinogenic factors, both domestic and professional. 30% of cancer is associated with exposure to chemical carcinogens and the authors did not analyze chemical carcinogens at all. Thus, the application of this approach is limited to the assessment of sensitivity to radiation carcinogenesis.

Ok We agree. There are a huge number of carcinogenic agents different from ionizing radiation. But the reviewer is aware that it is practically impossible to organize a paper with 5-10 different techniques and hundreds of carcinogenic agents. Anyway, the goal of this study is not the applicability to other carcinogenic agents but to verify whether two major carcinogenesis factors like mutability and control of cell cycle are sufficiently predominant to quantify cancer risk expressed as ERR, although aware that carcinogenesis is a multi-factorial process. Besides, the ERR and the majority of the assays applied here do not concern radiation (eg pPTSN plasmid prmits the measure of spontaneous hyper-recombination). See modified text in abstract and in Introduction.

  1. The Cancer Genome Atlas program showed huge number of genes involved, regulating not only recombination and cell proliferation, that in concordance with the publications of Professors Hanahan and Weinberg perfectly describing hallmarks of cancer. Thus, the information, which could be get using proposed approach is rather limited, also some correlation between cancer proneness and the intensity of DNA repair via homologous recombination and activity of cell proliferation may be observed.

OK: first, the first remark about limitation is somewhat unfair since to our knowledge, no report in literature provides data on all the genes of the cancer genome atlas, on both functional and quantitative features (eg vs ERR). Indeed, we are aware of the limitation of this study (only 8 cancer syndromes), and overall we did not hide such limitation in our previous version (see previous abstracts and conclusion). We have simply chosen the major cancer syndromes in terms of frequency and documentation. See modified in Introduction Discussion and Conclusions.

Second, there is a big confusion between homologous recombination, a DNA strand breaks repair pathway active in G2/M only and the hyper-recombination process, independent of the cell cycle. It must be stressed here that the cells tested are in majority in G0/G1 and provide hyperrecombination with all the assays: the minority of S-G2/M cells cannot explain the spontaneous or the radiation-induced response of all the cells seeded. To the contrary, in our conditions, we observed significant correlations and, overall, system of differential equations that cannot be the result of randomized processes, which suggests that our working hypotheses are not irrelevant. See modified text in Discussion

  1. The data revealing the association between cancer proneness and the intensity of DNA repair via homologous recombination and activity of cell proliferation should be presented more precisely. In the Result section authors demonstrate only some samples of the analyzed curves without detailed description how they were processed and integral data obtained. It is necessary to add tables including all the primary cell cultures analyzed, which could demonstrate final results of every analysis performed, so that to make possible to see the associations

First, we stress again on the confusion between homologous recombination and hyper-recombination. The reviewer should be aware that the probability that cancer can be initiated on quiescent cells is high, whatever the stress inducer.

Again, the analysis of data has been very fine since, to our knowledge, this is the first time that a system of differential equations is proposed for the initiation and promotion of cancer. We have added table in Supplementary data with all the numerical data. Furthermore, we have modified all the curves by adding the corresponding syndrome at each plot. See modified text and Figures in results and discussion.

  1. There is one more limit of the approach proposed in the study: it is performed using the cell of only one histogenesis type and radiation caused mainly blood malignancies, thus, in the analysis of cancer proneness, it is necessary to use cells of different histogenesis.

Mathematically, the robustness of our analysis did require a diversity in the syndromes and in the type of cancer to avoid any specificities and artefactual biases.

However, ataxia telangiectasia, Nijmegen syndrome, Fanconi Anemia, Bloom syndrome Li Fraumeni and neurofibromatosis type 1 i.e 6 among 8 syndromes (75%) are associated with high risk of lymphoma. However, these syndromes follow the natural high frequency of malignancies of blood by comparison to malignancies of other tissues. It is noteworthy that blood malignancies risk are also predicted by skin fibroblasts.

However, this proposition of the reviewer does not follow the ethical standards since some cells like brain cells cannot be sampled on living humans. Therefore, it will be impossible to consider brain tumor risk by studying human brain cells. Furthermore, the approach proposed by the reviewer will raise also the problem of extrapolation between tissues, which does not fall into the scope of our paper. Our approach is very different and does not follow the principle “the same histology type to predict the same tumor risk” proposed by the reviewer. Finally, how to proceed when a syndrome (like Li-Fraumeni) is at multi-tumor risks. No intercomparison will be possible. 

 To the contrary, the cutaneous fibroblasts and fibroblasts in general represent the majority of the conjunctive tissues, i.e. the most frequent histology type. Skin fibroblasts appeared therefore as the best compromise fro the following principle “ the same tissue to study different cancer risks” See modified text.

  1. There are many terminological and conceptual inaccuracies in the text of the manuscript, which are described in minor comments.

 Ok see our replies to your comments

Minor comments:

  1. Mutagenicity is a property of genotoxic agents/factors to induce DNA alterations, thus, it should not be used as a property of DNA. In the title and in the text “DNA mutagenicity” should be corrected to “DNA mutability”.

Yes you are right and we are fully agree. We apologize for such error. See modified text in all the manuscript.

  1. The phrase “Cancer proneness is associated with molecular hyper-recombination and cellular proliferation” should be corrected as Cancer proneness is associated with many factors, the line of which is very wide, and here it sounds as hyper-recombination and cellular proliferation are the only properties determining cancer proneness. In the context of the proposed methodology the idea of the phrase should be alike this one: “Activities of recombination and cellular proliferation are associated with radiation induced cancer proneness”.

OK We agree that there are many other factors contributing to cancer but 1°) each factor contributions are unequal; 2°) to choose at least one assay for each carcinogenic factor is not a reasonable and constructive approach. Hence, notwithstanding the fact that cancer proneness is associated with many factors, we made the hypothesis that the contributions of hyperrecombination and mutability and proliferation are major. The resulting correlations are consistent with such hypothesis. See also our replies below. Again, we do not necessarily focus on radiocarcinogenesis. See modified in Short Summary and in abstract and introduction

  1. The phrase “etiology of cancer remains poorly understood” is not correct, nowadays carcinogenic factors are well known and key characteristics of carcinogens are well described in the documents of the International Agency for Research on Cancer, Lyon, WHO. This agency elaborated perfect classification of carcinogenic factor of different nature (chemical, biological agents and physical factors). Modern concept of carcinogenesis is well-presented in the whole number of studies, showing simultaneous influence of multiple genotoxic and non-genotoxic promoting factors both on tumor appearance and tumor progression. Hallmarks of cancer, presenting the main peculiar characteristic of cancer cells, are also well-presented in the works of Prof. Weinberg and Prof. Hanahan.

Ok, you are right. We have deleted the sentence. However, despite the considerable efforts worldwide and notably those of IARC, the reviewer should admit that the IARC scales with the group « possibly carcinogenic » and « probably carcinogenic » suggest that further investigations are needed. Furthermore, the IARC classification is not quantitative and, for example, does not integrate dose-response or concentration-response curves. Hence, there is still no quantitative direct link between carcinogenic factor and ERR. As a consequence, there is no intercomparison between the biological effects of carcinogenic factors. See modified text in Introduction

  1. The phrase “Molecular investigations have paralleled the hypotheses of the initiation, promotion and progression steps with the identification of some tumor suppressor genes, called caretakers when responsible for genome surveillance, and gatekeepers, when responsible for control of the cell cycle checkpoints” requires serious revision. Initiation, promotion and progression are the steps of carcinogenesis, which are well documented, it is not the hypothesis, concerning the genes, involved in carcinogenesis. Exposure of cells to genotoxic compounds, leading to mutation appearance in many genes involved in carcinogenesis, corresponds to cancer initiation, and regulation of these gene activity by non-genotoxic carcinogens, corresponds to the process of promotion. As concerns the genes involved into carcinogenesis, they may be separated into different groups in dependence on their function. Thus, in parallel may be considered classifications of the genes involved into carcinogenesis to (1) oncogenes and genes of tumor suppressors or (2) caretakers and gatekeepers. Although, nowadays, the publications of Hanahan and Weinberg determined more popular classification of the genes, involved into carcinogenesis, dependently of their involvement in the different processes corresponding to Hallmarks of cancer. Cancer progression occurs as the result of the influence of both genotoxic and non-genotoxic agents/factors. A very interesting results was obtained by Atlas of cancer genome program showing that only a small number of genes are not involved in carcinogenesis.

Recently, we have provided a review about the epistemology of cancer models that shows clearly that there was a period of confusion with regard the definition of these terms. For example, the reviewer should admit that BRCA1 gene was considered as a tumor suppressor gene (eg Galli et al.Int J Mol Sci 2024) but also as a gatekeeper (eq Zou et al. Cell Death Dis, 2018). The same for ATM, p53, … and even all the genes represented by the syndromes tested here. See modified text in Introduction

We do not understand the purpose of the last sentence that does not fall in the scope of our paper and does not propose a rational approach to better understand carcinogenesis.

  1. Proneness to cancer depends on the enzyme system of carcinogen activation and detoxification and all the enzymes of DNA repair, antitumor immunity, epigenetic regulation and other types of DNA transcription activation. Complex composition of polymorphic variants of all these gene, which function under influence of different environmental factors, determine the individual risk of cancer (or cancer proneness). Is it reasonable to determine just two factors influenced the process, even if these factors are important?

At least, t is necessary to present the goal of the study focusing on carcinogenesis induced by radiation. It is also very important, in particular, for people working with professional radiation exposure and for patients treated with radiotherapy.

Again, the goal of our investigations was inverse: among the numerous factors involved in the carcinogenesis process, what are the most representative to provide predictive assays. The pioneer work of the D Scott ‘s team proves that It is possible to propose an assay, predictive of cancer proneness that is not linked directly to hypoxia, immunity, methylation, …See modified text in Introduction.

 If such assay accounts for “spontaneous” cancer proneness, does it account for the proneness of “induced” cancer. Here, by considering only mutability and proliferation capacities, the existence of correlations with ERR and of a system of differential equations with functions of ERR provide a proof that our approach was not so irrelevant.

Hence, we agree with the reviewer about the fact that carcinogenesis is a multi-factorial process but some factors or combination of factors may be sufficiently representative to predict cancer proneness. This hypothesis is not new: the team of David Scott in Manchester has developed G2 assay from cytogenetic that was able to predict cancer proneness. It is because this cytogenetics assay was too time consuming that we tried to develop another (binary) approach.

See modified text in Introduction and Discussion

  1. The phrase “With regard to cell cycle control assays, three major categories exist: those assessing the G1/S transition (like BrdU/EdU incorporation assay), those evaluating the G2/M arrest (like mitotic index assay), and finally flow cytometry that encompasses all the stages of cell cycle” requires more precise explanation.

Ok, However there are only 3 arrests possible (G1, G2/M and S). See modified text and notably the Table S1 that summarizes all the numerical values.

  1. The authors pointed out that “More than 50 nuclei were analyzed per experiment per post-irradiation time, with three independent replicates performed”. It should be pointed out how it corresponds to OECD guidelines.

As already written in our last version, the foci scoring procedure was performed manually with a series of verifications inter-lab and inter-reader and has received certification under CE mark and ISO-13485 quality management system norms [28,33], which is much more constrained than the GPL and OCDE Guidelines. Again, it must be stressed that the foci scoring procedure was applied in the frame of the Soleau Envelop and patents (FR3017625 A1, FR3045071 A1, EP3108252 A1).

  1. The data obtained should be presented in more details to see what syndrome causes what change of the property, please, provide the corresponding information in details in tables in the main text.

Ok see modified text, new table S1 and the name of syndromes in all the graphs.

Reviewer 3 Report

Comments and Suggestions for Authors

This study examined how mutations and cell proliferation influence cancer risk by analyzing skin cells from individuals with eight major genetic cancer syndromes. The authors used various assays to measure cell mutagenicity, such as plasmid assays and mutation frequency tests and assessed cell proliferation using flow cytometry. They found notable correlations between DNA mutation rates, cell cycle arrest, and cancer risk. The researchers developed mathematical models that describe these relationships and discovered that combining mutation rates with cell proliferation data can more accurately predict cancer risk. These results suggest that a better understanding of these cellular processes could improve cancer risk assessments and highlight the need for further research with additional genetic syndromes. 

Some of my minor concerns.

1. Could you explain each method in this work such as recombination plasmid assay, HPRT and MRE11 assay in a better way and elaborate on the function, their advantages and disadvantages? This is to engage broader audiences. 

2. Could you explicitly mention the function of HPGT and the role of TG in the assay?

3. Please ensure consistency in font usage and paragraph formatting throughout, including for formulas.

Curiosity questions.

1. Could you provide more details on how you measured the kinetics of MRE11 foci after irradiation? What methods were used to determine the peak of MRE11 nuclease activity, and how were these measurements standardized across experiments?

2. You found no correlation between MRE11max and HPRT mutation frequency or FDR. Can you suggest potential reasons for this lack of correlation and how it might influence the interpretation of MRE11 activity concerning hyper-recombination and genomic instability?

3. With the 15% systematic relative error range for ERR values, could you explain how this margin was determined and its impact on the reliability of your cancer risk conclusions for the studied syndromes?

4. Your analysis showed sigmoidal and curvilinear relationships between hyper-recombination, HPRT mutations, and ERR. Could you discuss the biological significance of these relationships and how they help in understanding cancer susceptibility in these genetic syndromes?

5. How do your findings fit within the RIANS model for radiobiological features and cancer risk? Are there aspects of the model that may need adjustment based on your results?

6.  Are there specific genetic syndromes or experimental methods you will focus on to enhance cancer risk prediction? What are your plans for future research to further validate and expand on this study findings?

7. You observed a strong correlation between hyper-recombination rates and cell proliferation capacity in predicting cancer risk. Can you explain the underlying mechanisms of this correlation and its potential implications for cancer risk assessment and treatment strategies?

This research work is fresh and examines how mutations and cell proliferation influence cancer risk by analyzing skin cells. The researchers developed mathematical models that describe these relationships and discovered that combining mutation rates with cell proliferation data can more accurately predict cancer risk. These results suggest that a better understanding of these cellular processes could improve cancer risk assessments and highlight the need for further research with additional genetic syndromes. I find the work interesting and useful to the cancer biology community. I recommend this work for publication in cancers with minor edits. Congratulations to the authors.

Comments on the Quality of English Language

I recommend that the authors undertake an additional round of proofreading and focus on improving the quality of the figures, especially Figure 8

Author Response

Reviewer 3 :

We thank the reviewer 3 for his/her comments

This study examined how mutations and cell proliferation influence cancer risk by analyzing skin cells from individuals with eight major genetic cancer syndromes. The authors used various assays to measure cell mutagenicity, such as plasmid assays and mutation frequency tests and assessed cell proliferation using flow cytometry. They found notable correlations between DNA mutation rates, cell cycle arrest, and cancer risk. The researchers developed mathematical models that describe these relationships and discovered that combining mutation rates with cell proliferation data can more accurately predict cancer risk. These results suggest that a better understanding of these cellular processes could improve cancer risk assessments and highlight the need for further research with additional genetic syndromes. 

  • Some of my minor concerns.

  1. Could you explain each method in this work such as recombination plasmid assay, HPRT and MRE11 assay in a better way and elaborate on the function, their advantages and disadvantages? This is to engage broader audiences.

See modified text in introduction and in discussion

  1. Could you explicitly mention the function of HPGT and the role of TG in the assay?

OK See modified text in Introduction.

  1. Please ensure consistency in font usage and paragraph formatting throughout, including for formulas.

OK you are right. We do not know the reason of these typos. We apologize. See modified text

  • Curiosity questions.

  1. Could you provide more details on how you measured the kinetics of MRE11 foci after irradiation? What methods were used to determine the peak of MRE11 nuclease activity, and how were these measurements standardized across experiments?

The data time plots were fixed to 0, 5 min,10 min, 30 min, 1h, 4h et 24h. The MRE11max was deduced from the experimental data and not by calculation of the potential maximum after data fitting.. See also materials and methods and results.

  1. You found no correlation between MRE11max and HPRT mutation frequency or FDR. Can you suggest potential reasons for this lack of correlation and how it might influence the interpretation of MRE11 activity concerning hyper-recombination and genomic instability?

OK we have already suggested that such absence of correlation may indicate that MRE11 is not necessarily involved in the HPRT mutation production and in the whole nuclease process. See modified text in Results and Discussion/

  1. With the 15% systematic relative error range for ERR values, could you explain how this margin was determined and its impact on the reliability of your cancer risk conclusions for the studied syndromes?

After reviewing literature about ERR, the average ERR was associated to relative error of less than 15% : see modified text in Materials and Methods and discussion

  1. Your analysis showed sigmoidal and curvilinear relationships between hyper-recombination, HPRT mutations, and ERR. Could you discuss the biological significance of these relationships and how they help in understanding cancer susceptibility in these genetic syndromes?

OK  We have developed again the discussion about the data interpretation. See modified  text in disucssion.

  1. How do your findings fit within the RIANS model for radiobiological features and cancer risk? Are there aspects of the model that may need adjustment based on your results?

OK  We have developed again the discussion about the link between RIANS model. See modified text in Discussion

  1. Are there specific genetic syndromes or experimental methods you will focus on to enhance cancer risk prediction? What are your plans for future research to further validate and expand on this study findings?

OK See modified text in Conclusion.

  1. You observed a strong correlation between hyper-recombination rates and cell proliferation capacity in predicting cancer risk. Can you explain the underlying mechanisms of this correlation and its potential implications for cancer risk assessment and treatment strategies?

OK  We have developed again the discussion about the H x G product

Round 2

Reviewer 1 Report

Comments and Suggestions for Authors

The quality of the manuscript has been improved. However, some points are still not fixed. 

Example 1: Graphical abstract was not added at the end of the introduction. 

Example 2: Aim of the work was not explained at the end of the introduction. 

Author Response

Reviewer 1 : 

We thank the reviewer for his/her comments

The quality of the manuscript has been improved. However, some points are still not fixed. 

Example 1: Graphical abstract was not added at the end of the introduction. 

OK see modified text and the introduction of graphical abstract in the text

Example 2: Aim of the work was not explained at the end of the introduction.

OK see modified text at the end of the introduction

Reviewer 2 Report

Comments and Suggestions for Authors

I really appreciate the huge and well-done work of the authors with the manuscript. However, X-rays mainly cause DNA strand breaks. Inhibition of DNA strand break repair may have indirect consequences on point mutation repair, as non-repaired lesions finally could be followed by strand break appearance.   Authors analyze just the only cell treatment with X-rays causing strand breaks and made a conclusion on cancer proneness on the whole, thus, including cancer proneness to genotoxic carcinogens. Point mutations represent consequences of modified bases, DNA adducts and upurine/upirimidine sites. Direct association may not correspond to the indirect one. Thus, I would propose to enter in the title of the study this detail: Prediction of cancer proneness under influence of X-rays with DNA mutability and/or 3 cellular proliferation assays. The response of the authors that it is difficult to organize the study including bot X-ray and point mutation inductor treatment should not be followed by ignoring of the fact described above. Moreover, it would be desirable to point out that authors describe just an approach, not a test, as the test validation require determination of prognostic significance, calculated on the basis of right and false positive and negative predictions and it is very difficult to organize.

Thus, the manuscript can be published after entering the described changes in the title or, at least, discussion of this point by the Cancers Editorial Board.

Author Response

Reviewer 2

We thank the reviewer 2 for his/her comments.

I really appreciate the huge and well-done work of the authors with the manuscript. However, X-rays mainly cause DNA strand breaks. Inhibition of DNA strand break repair may have indirect consequences on point mutation repair, as non-repaired lesions finally could be followed by strand break appearance.   Authors analyze just the only cell treatment with X-rays causing strand breaks and made a conclusion on cancer proneness on the whole, thus, including cancer proneness to genotoxic carcinogens. Point mutations represent consequences of modified bases, DNA adducts and upurine/upirimidine sites. Direct association may not correspond to the indirect one. Thus, I would propose to enter in the title of the study this detail: Prediction of cancer proneness under influence of X-rays with DNA mutability and/or 3 cellular proliferation assays. The response of the authors that it is difficult to organize the study including bot X-ray and point mutation inductor treatment should not be followed by ignoring of the fact described above. Moreover, it would be desirable to point out that authors describe just an approach, not a test, as the test validation require determination of prognostic significance, calculated on the basis of right and false positive and negative predictions and it is very difficult to organize.

Thus, the manuscript can be published after entering the described changes in the title or, at least, discussion of this point by the Cancers Editorial Board.

We fully agree to modify the title accordingly but we propose to complete it again by the number of assays applied for mutability and not only mentioning the number of proliferation assays because the MRE11 assay may be under the influence of X-rays . Therefore  we propose : 

"Prediction of cancer proneness under influence of X-rays with 4 DNA mutability and/or 3 cellular proliferation assays" See modified title

Round 3

Reviewer 1 Report

Comments and Suggestions for Authors

The manuscript improved well.